

# Bonsai-SPH: A GPU accelerated astrophysical Smoothed Particle Hydrodynamics code

Jeroen Bédorf[1,2]* and Simon Portegies Zwart[1]

**1** Leiden Observatory, Leiden University, P.O. Box 9513, 2300 RA Leiden, the Netherlands
**2** Minds.ai, Inc., Santa Cruz, the United States

* bedorf@strw.leidenuniv.nl

## Abstract

We present the smoothed-particle hydrodynamics simulation code, BONSAI-SPH, which is a continuation of our previously developed gravity-only hierarchical $N$-body code (called BONSAI). The code is optimized for Graphics Processing Unit (GPU) accelerators which enables researchers to take advantage of these powerful computational resources. BONSAI-SPH produces simulation results comparable with state-of-the-art, CPU based, codes, but using an order of magnitude less computation time. The code is freely available online and the details are described in this work.

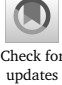

# 1  Introduction

Smoothed Particle Hydrodynamics (SPH) is a particle based simulation method for fluids. The method was introduced in astrophysics in 1977 by [1] and [2], and since then used in numerous simulations of astrophysical phenomena addressing a wide range of phenomena including (but far from complete) processes such as supernovae feedback [3], supernova shell morphology [4], galaxy dynamics [5], evolution of disks around and accretion onto black holes [6,7], star formation [8,9], mass-transfer in binary and triple-star systems [10], stellar evolution [11], stellar collisions [12,13], and planet formation [14,15].

There are several different SPH prescriptions, each with its own specific advantages and disadvantages. Many of the slightly different implementations of generally the same algorithm are available in the public scientific domain, or private. Well known public implementations include TreeSPH [16], Phantom [17], and Gadget [18,19]. Here, we present, yet another public SPH code. The main difference between other codes and our implementation is that we use Graphical Processing Units to accelerate the calculations. This enables a faster execution compared to existing codes and therefore the simulation of larger models within the same wall-clock time-frame. We deliberately do not develop another SPH prescription, but take advantage of the large body of previous work as described in recent reviews [20–22] and references therein.

SPH is one of several commonly used methods to simulate fluid dynamics, another well known group of method are the Eulerian (mesh/grid-based) solvers. Each method has its own advantages and disadvantages. For example, one SPH's key strengths is that it is self-adaptive, because it is based on particles. With mesh-based methods one has to resort to adaptive meshes in order to achieve a similar level of flexibility and computational efficiency. The result is a limited domain range for models that contain low density regions. A commonly cited disadvantage of SPH is the method's difficulty with capturing shocks. This is because the method has inherent zero intrinsic dissipation and one therefore has to add dissipative terms such as artificial viscosity [23]. On the other hand shocks are handled naturally in grid-based methods [24]. However, when the proper viscosity settings are chosen SPH is capable of handling shocks, such as those occurring in the Kelvin-Helmholtz instability test [17].

Recently there have been various methods introduced that combine the best parts of SPH and mesh based methods and combine this into a so called moving-mesh method. In this method the grid cells move with the fluid flow [25–28]. Although these methods are computationally intensive none of them are accelerated by GPUs although some work is being done on getting the underlying methods to run on GPU processors [29].

Given the large amount of literature available that discuss the advantages and disadvantage of the various hydrodynamic methods we refer the reader to the following excellent reviews but point the interested reader to the references in this work, in particular to [20,21,23,30,31] and do not go into further detail here unless it directly touches upon the goals of this paper,

accelerating the SPH method with the use of Graphical Processing Units.

Most of the previous work related to hydrodynamics and GPUs has focused on mesh-based codes. For example, in a version of FLASH [32, 33] the authors accelerate the (non-hydrodynamic) gravity computation using GPUs. The optimization of the gravity module helps speeding up the code, but the hydrodynamics modules are still running on the CPU and form a major fraction of the total compute time. Another approach is taken in GAMER [34, 35], here the authors implement the grid based hydro computations on the GPU and present results that are qualitatively comparable to FLASH, but using over an order of magnitude less compute time. Previous work directly related to GPU accelerated astrophysical SPH methods is limited, however there is previous work done on non-astrophysical SPH methods. For example GPUSPH [36] and DualSPHysics [37], which are CFD codes that use the SPH algorithm.

This lack of available codes and the fact that in SPH the organization of the particles can be done efficiently with the use of hierarchical data-structures (trees) [16,18] has motivated us to develop a new optimized SPH code. We build this new SPH code on top of our existing gravity only simulation code BONSAI [38, 39], which uses an hierarchical data-structure to compute gravity and as such can naturally be extended to simulate fluid dynamics. BONSAI-SPH has been developed to take advantage of Graphics Processing Units (GPUs) accelerators and all the actions related to the tree are executed on the GPU. This results in a high performance, scalable, simulation code that enables us to perform the simulation of very high-resolution models in reasonable time [40]. With the increased availability of GPUs in the world's largest supercomputers [41] it will allow researchers to take advantage of this new infrastructure and as such perform larger simulations than possible before.

This work is organized as follows, in Sect. 2 we shortly introduce the SPH method and the specific version of SPH that we use in this work, in Sect. 3 we describe our implementation, in Sect. 4 we present the results and in Sect. 5 we present our conclusion and suggestions for future work.

## 2 Smoothed Particle Hydrodynamics

### 2.1 Overview

In SPH the simulated fluid is discretised into a set of particles where each particle has a position, $p$, velocity, $v$, and mass, $m$. This enables SPH to solve the hydrodynamics equations in the Lagrangian form. In contrast a grid code, as for example, ENZO [42] solves the hydrodynamical equations using a Eulerian form. Where the difference is that in the Lagrangian form you change the properties of the individual particles as the fluid moves. While in the Eulerian form you change the properties of fixed locations as the fluid passes through these locations. More information about the difference between these two methods as used in astrophysics can be found in [24] and references therein. See also [43] for a comparison between the various methods.

In SPH the properties of a particle are based on its nearest neighbours. The contribution of each neighbour depends on the distance between the particle and the neighbour, where further away neighbours contribute less than nearby neighbours. For example, the density of a particle is computed by the sum over neighbouring particles that fall within the smoothing length distance from the particle. The smoothing length acts as the search radius and depends on the number of nearby neighbours. Particles that are located outside this radius do not contribute to a particle's density. This process is illustrated in Fig. 1. Here the particle that we target is drawn in the center and around it we have a circle with radius $h$. The kernel indicates the strength of the neighbour contribution where from particles near our target the

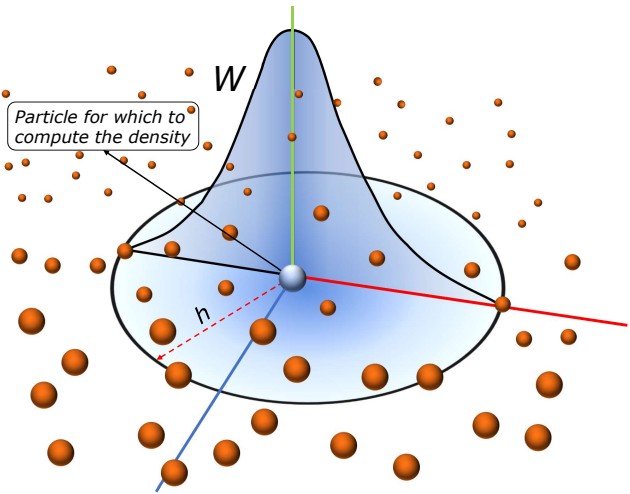

Figure 1: Illustration on how the SPH kernel functions. All particles within radius $h$ contribute a fraction to the density of the target particle. The fraction is related to the distance, the closer the more contribution which is illustrated by the height of the density curve.

contribution is higher (peak in the distribution) and particles that fall outside $h$ contribute nothing. The exact shape of kernel $W$ depends on the chosen kernel which we discuss in section 2.3.

## 2.2 Fundamental equations

In order to accurately discretise the continuous fluid space into discrete particles each particle has to be associated with a density. The density of a particle is computed via,

$$\rho_i = \sum_{j=1}^{N} m_j W(|\mathbf{r}_{ij}|, h_i), \tag{1}$$

$\mathbf{r}_{ij} = r_i - r_j$, $W(r, h)$ is the smoothing kernel and $h_i$ the smoothing length of an individual particle. The Smoothing length relates to the particle's density and mass via,

$$h_i = h_{\text{fact}} n_i^{-1/3} = h_{\text{fact}} (\frac{m_i}{\rho_i})^{1/3}. \tag{2}$$

Here $n$ is the particle number density, $\rho$ the density, $m$ the mass, and $h_{\text{fact}}$ a proportional factor that is specific to the used density kernel [21].

In BONSAI-SPH we implemented the same SPH equations as used by PHANTOM [17] and as such make use of the gradient based SPH method, we therefore compute the density gradient,

$$\frac{d\rho_i}{dt} = \frac{1}{\Omega_i} \sum_{N}^{j} m_j (v_i - v_j) \cdot \nabla i W_{ij}(h_i)), \tag{3}$$

$W_{ij}(h_i) \equiv W(|r_i - r_j|, h_i)$ is the contribution of the smoothing kernel between particle $i$ and $j$ given smoothing length $h_i$. This is a function of the gradient of the smoothing length,

$$\Omega_i \equiv 1 - \frac{\partial h_i}{\partial \rho_i} \sum_j^N mj \frac{\partial W_{ij}(h_i)}{\partial h_i}. \tag{4}$$

For details on how these equations are derived see [44, 45].

The above equations are independent of the used smoothing kernel, with the only requirement that it is differentiable.

## 2.3 Smoothing kernels

There are three smoothing kernels implemented in BONSAI-SPH. Depending on the goal of the simulation the user can select, at compile time, which of these kernels is the most suitable. The reason for doing this at compile time is to improve the efficiency, since a number of the kernel operations involve constants and as such can be optimized by the compiler.

The following kernels are available:

- $M_4$ cubic spline kernel, a kernel based on the B-spline family [46] and the most commonly used SPH kernel [47].
- $M_6$ quintic kernel, this is a higher order extension of the $M_4$ kernel which requires a larger smoothing range, and therefore more neighbours which makes it computationally more expensive.
- Wendland $C^6$, this is one of the more recently developed kernels and proved to be stable against the common pairing instability problem [48, 49].

The choice of kernel is an open discussion with no forgone conclusion, see for example the discussions in [21, 48].

## 2.4 Time-integration

For the time integration we use the same second order Leapfrog integrator [50] as used in BONSAI. In this scheme the position and velocity are predicted to the next simulation time using previously calculated forces. Then the new densities and forces are computed after which the velocities are corrected. This is done for all particles in parallel using the globally determined minimum time-step[1]. This process is described in equations 5 to 7.

$$r_1 = r_0 + v_0 \delta t + \frac{1}{2} a_0 (\delta t)^2, \tag{5}$$

$$v_{1p} = v_0 + a_0 \delta t. \tag{6}$$

Next the densities and forces are computed after which the velocity undergoes the correction step,

$$v_{1c} = v_{1p} + \frac{1}{2}(a_1 - a_0)\delta t, \tag{7}$$

where $a$ is the acceleration (see Eq. 9).

The time-step is determined after each iteration and constrained by the Courant time-step [51],

$$\delta t^i \equiv C_{\text{cour}} \frac{h_i}{v_{sigmax,i}^{dt}}. \tag{8}$$

We use $C_{\text{cour}} = 0.3$ following [51], and $v_{sigmax,i}^{dt}$ is the maximum signal speed (see Eq. 12) between particle $i$ and its neighbours.

The acceleration, $a$, used in Eqs. 5-7 is defined as the sum over all the contributing neighbours $j$:

---

[1]For this work we make no use of the block-time step capabilities of the code.

$$a = -\sum_j m_j \left[ \frac{P_i + q_{ij}^i}{\rho_i^2 \Omega_i} \nabla_i W_{ij}(h_i) + \frac{P_j + q_{ij}^j}{\rho_j^2 \Omega_j} \nabla_i W_{ij}(h_j) \right] + a_{\text{grav}}^i. \tag{9}$$

Here $q_{ij}^i$ and $q_{ij}^j$ are the artificial viscosity terms (see below) and $a_{\text{grav}}^i$ is the Newtonian force exerted on particle $i$. Note that the Newtonian force is a contribution by all particles and not just the neighbours within the smoothing range. In our work the gravitational force is computed using the Barnes-Hut hierarchical tree algorithm [52].

The last property computed during each iteration is the internal energy which discretised form is given by,

$$\frac{du_i}{dt} = \frac{P_i}{\rho_i^2 \Omega_i} \sum_j m_j \mathbf{v}_{ij} \cdot \nabla_i W_{ij}(h_i) + \Lambda_{\text{shock}}. \tag{10}$$

Here $\Lambda_{\text{shock}}$ is the artificial conductivity shock capturing term, discussed below.

## 2.5 Artificial viscosity

The artificial viscosity terms control the dissipation of the shock-capturing equations for the equations of motion [53]. The switch is defined via,

$$q_{ij}^i = \begin{cases} -\frac{1}{2} \rho_i v_{\text{sig},i} \mathbf{v}_{ij} \cdot \hat{\mathbf{r}}_{ij}, & \mathbf{v}_{ij} \cdot \hat{\mathbf{r}}_{ij} < 0. \\ 0 & \text{otherwise,} \end{cases} \tag{11}$$

with $\hat{\mathbf{r}}_{ij} \equiv (\mathbf{r}_i - \mathbf{r}_j)/|\mathbf{r}_i - \mathbf{r}_j|$ and $\mathbf{v}_{ij} \equiv \mathbf{v}_i - \mathbf{v}_j$ which form the unit vector between the particles and $v_{\text{sig}}$ is the signal speed, given by

$$v_{\text{sig},i} \equiv \alpha^{AV} c_{s,i} + \beta^{AV} |\mathbf{v}_{ij} \cdot \hat{\mathbf{r}}_{ij}|, \tag{12}$$

where $\alpha^{AV}$ and $\beta^{AV}$ are configuration parameters that influence how the shocks are treated. In BONSAI-SPH they are set at the start of the simulation and are not updated overtime. This is different from PHANTOM which uses a more sophisticated control switch that can update $\alpha^{AV}$, per particle, during the simulation. More details and discussions on how to set these parameters can be found in [51, 54].

## 2.6 Artificial conductivity

As mentioned above to handle shocks in the computation of the internal energy there is oftentimes a viscosity parameter added. In BONSAI-SPH we follow [17] and implemented a conductivity term. The combination of both the artificial viscosity and artificial conductivity is defined via,

$$\Lambda_{\text{shock}} \equiv -\frac{1}{\Omega_i} \sum_j m_j v_{\text{sig},i} \frac{1}{2} (\mathbf{v}_{ij} \cdot \hat{\mathbf{r}}_{ij})^2 F_{ij}(h_i) + \sum_j m_j \alpha_u v_{\text{sig}}^u (u_i - u_j) \frac{F_{ij}(h_i)}{\Omega_i \rho_i} + \frac{F_{ij}(h_j)}{\Omega_j \rho_j}. \tag{13}$$

Here $\alpha_u$ is the configurable parameter that controls the strength of the thermal conductivity. In general, we keep this fixed to $\alpha_u = 1$. The force $F_{ij}$ is defined via,

$$F_{ij} \equiv \frac{C_{\text{norm}}}{h^4} f'(q). \tag{14}$$

Here $C_{\text{norm}}$ is a property of the chosen smoothing kernel and $f'(q)$ the derivative of $q = (|r_i - r_j|/h)$.

Finally, the signal speed, $v_{\text{sig}}^u$, is defined via

$$v_{\text{sig}}^u = \sqrt{\frac{|P_i - P_j|}{\bar{\rho}_{ij}}}, \tag{15}$$

where $P$ is the pressure of a particle and $\bar{\rho}_{ij}$ the average density of particles $i$ and $j$.

## 3 Implementation

This section starts with a short overview of BONSAI followed by a short description of the most important algorithms and how they are implemented on the GPU. For each of these algorithm we describe the differences between BONSAI and BONSAI-SPH to indicate how adding support for fluid dynamics affects the basic tree related algorithms. A full description on how BONSAI works can be found in [38, 39] or by inspecting the source-code in the online repository [55].

The development of BONSAI started in 2010 with the goal to develop an $N$-body code that was, from the ground up, optimized for GPU accelerators. When the code was completed the full software stack was executed on the GPU. This eliminated data transfer requirements and allowed the $\mathcal{O}(N)$ parts of the code to take advantage of the additional processing and bandwidth capabilities of the GPU. When targeting a single GPU, the CPU's tasks are limited to basic operations and orchestrating compute work on the GPU. This frees up the CPU cores for other work, such as on-the-fly post-processing or visualizations. When executing BONSAI in a distributed setup the CPU is responsible for handling (network) communication between the GPUs. The communication patterns depend on the number of nodes and the simulated model, exactly the kind of irregular tasks for which the CPU is perfectly suited. The distribution of work between the CPU and GPU, is such that each can focus on its strengths, and allows BONSAI to scale to thousands of GPUs while maintaining high computational efficiency.

For the SPH additions we stick to the above design pattern and keep all the compute work and data storage on the GPU. In practice this means that additional memory is reserved for storing the fluid dynamic properties such as pressure, density, energy, etc. Furthermore, compute kernels are added to compute hydro properties.

The GPU portions of the software are developed using CUDA which in practice means that only NVIDIA GPU hardware is supported. BONSAI-SPH works on the Tesla, GeForce and Quadro GPU series as long as the compute capability of the device is 3.0 or greater.

### 3.1 Tree Construction

The tree-code algorithm is based on the assumption that particles can be grouped in a hierarchical data-structure. Most CPU based algorithms create an octree data structure by performing sequential particle insertion which adds particles to a box that encloses the spatial coordinates of the particles. Once the box is full the box is split up into 8 sub-boxes (hence the name octree) and the particles that were in the original box are divided over those sub-boxes. For GPU based algorithms this is not efficient as there would be too many race conditions when multiple threads become involved. Therefore, we use a different kind of method that can be executed by many threads in parallel. The method uses a space filling curve [56] to order particles into the boxes. Each particle is assigned a unique location on the curve, based on the coordinates of the particle. Next, the particles are sorted such that their order in memory follows the space filling curve. Both these operations are executed on the GPU, where Thrust[2] is used to perform the sort operation. Next the octree is constructed by chopping the space

---

[2]https://developer.nvidia.com/thrust

filling curve into sections where each section refers to part of the tree. This way the tree is built level by level until the smallest section of non-chopped curve contains at most $N_{\text{leaf}}$ particles. Where $N_{\text{leaf}}$ stands for the maximum number of particles that is assigned to a leaf, an end point of a tree branch.

Using the same set of sorted particle we create *particle groups*. These particle groups are used during the tree-traverse, where a group of particles traverses the tree instead of individual particles. For the group construction we again chop the space filling curve into sections. The sections are chopped into smaller sections until each section contains at most $N_{\text{group}}$ particles.

The above described method is the same for both BONSAI and BONSAI-SPH with the difference that for BONSAI-SPH lower values are used for $N_{\text{leaf}}$ and $N_{\text{group}}$. The lower values give better performance during the tree-traverse required for computing SPH properties as discussed in the next section.

## 3.2  Tree-traverse for SPH

The tree-traverse for the density/hydro-force computations are similar to the method used for the gravity computation. However, where the gravitational force requires information from all particles, either via direct interaction or via multipole expansion approximations, the SPH method only requires information from particles that fall within the smoothing range (see Fig. 1). This has a number of consequences which we will list after giving a global description of the tree-traverse method.

CPU based tree-traverse algorithms are often implemented via a recursive algorithm. For the GPU processor this is not a good fit and instead we use a distributed breath first traversal algorithm. Furthermore, particles do not traverse the tree individually but in a group of particles, this is known as Barnes' vectorization [57] and improves the GPU utilization as groups of particles perform the same operations in parallel.

For our GPU implementation we make extensive use of in- and exclusive scan algorithms, for example to expand compressed node indices. Say, if a leaf contains 8 particles then we only store the index of the first particle, and the number of particles. This tuple, for example (104, 8), is then expanded as follows: 104, 105, 106, 107, 108, 109, 110, 111. To optimize the performance and reduce the amount of memory resources required, the scan algorithms are implemented with the use of shuffle instructions and embedded PTX code.

During the tree-traverse individual tree-nodes are tested and for each node is decided if it has to be expanded (traversed further) or that it falls outside of the search range. The possible options are,

- If a node falls outside the search range then it is either discarded (when computing fluid dynamic properties), or it is put on a *multipole approximation evaluation list* (gravity computation). The list is stored in the GPUs on chip shared memory and therefore has a relative limited size, but allows for quick access. Once the list is full it is processed in parallel by the threads traversing the tree and the multipole approximation between the tree-nodes and the particles that are part of the group traversing the tree is computed.
- If a node falls inside the search range and it is not a leaf then it will be added to the *next level list* and processed further during the next loop of the tree-traverse algorithm.
- If a node falls inside the search range and it is a leaf then the individual particles of the leaf are added to the *particle evaluation list*. Once the list is full the list is processed. This is described in more detail in Sect. 3.2.1.

The tree-traverse is continued until the next level list is empty which indicates that all relevant sections of the tree have been processed.

The useful or not decision as made during the tree-traverse is different for the gravity and fluid dynamic computations. For gravity the decision is based on the distance between a

particles group and a tree-node with respect to a specified *multipole acceptance criteria*. While for fluid dynamics it is based on the distance between a particle group and the tree-node and if this is less than the smoothing range of the particle group.
This has the following consequences for BONSAI-SPH

- The difference between interaction lists of individual particles is larger when compared to the gravity interaction lists. A too large a difference leads to executing unneeded computations. To reduce the difference we use a more fine-grained interaction path, achieved by using smaller values for $N_{\text{leaf}}$ and $N_{\text{group}}$ when computing fluid properties. This causes particle groups and leaf nodes to be physically smaller, and thereby reducing the number of non-useful interactions.
- For SPH there are two properties that are computed via tree-traverse operations, namely the density and the hydrodynamic force which both require a slightly modified tree-traverse. The difference lies in how it is decided which particles should interact. For the density computation this is determined via the smoothing range of the particle traversing the tree. For the hydro-force computation it is required, in order to preserve angular momentum, that a force is computed if one particle falls within the smoothing range of the other. Therefore we use the maximum smoothing of the two candidate particles to determine if the particle has to be added to the interaction list.

The lower values for $N_{\text{leaf}}$ and $N_{\text{group}}$ reduce the efficiency of the gravity computation, but this is offset by large efficiency gains for the density and hydro-force computations.

### 3.2.1 Interaction list processing

When processing the interaction list for the fluid dynamic computations, we either execute density or hydro-force computations. The function to execute is a templatized parameter of the tree-traverse code. Each thread of the group is responsible for loading part of the interaction list in memory and sharing it with neighbouring threads when the density and hydro-force functions are executed. After processing the list the partial results are returned and the tree-traverse continues.

As with the gravity computation these functions are optimized and make use of the shared memory and shuffle instructions available on NVIDIA GPUs via the CUDA language. Compared to the gravity computation the number of required resources is considerable larger when computing SPH related properties. This has a negative effect on the performance, because data will be flushed from registers to main memory. However, the overall performance is still better than when using CPUs (see Sect. 4).

During each time-step we run the density computation multiple times in order to let it converge on the correct number of neighbours. Currently this iteration is done 3 times for all particles, a future optimization would be to make the number of iterations dynamic with a per-particle group coarseness. The hydro-force computation is executed only once as there is no convergence requirement.

### 3.3 Particle integration

In order to move particles forward in time and update properties such as density and energy the newly computed forces have to be applied on the particles. For this a grid of compute threads is launched on the GPU. Each thread is responsible for processing a single particle. This method is the same for both the gravity and hydrodynamic version of the software.

### 3.4 Multi-GPU

The multi-GPU implementation is an extension of the gravity version, described in detail in [39], which uses the local essential tree (LET) method to exchange data with neighbouring processes [58]. In BONSAI the LET tree is built on the CPU, while the GPU is computing gravitational forces. The CPU then sends/receives the LET trees that contain the particle properties required to compute the total gravitational force exerted on a particle. For BONSAI-SPH the method is extended to, next to the properties required for gravity, include the tree-node and particle properties required for hydrodynamics. In addition we improved some of the existing pre-compute operations (e.g. detection of domain boundaries) by migrating them from the CPU to the GPU in order to reduce the amount of required memory copies.

In practice it turned out that the parallelization method did not work as efficiently for SPH as it does for gravity. The tree-traverse to create LET structures has to perform a deeper traversal to determine which particles are important. This is a consequence of having more fine grained groups (8 vs 64 particles per group). In practice this leads to increased CPU processing time which hinders the scalability and results in nearly no improvement in execution time. It is possible to tune this selection process further, either by setting a depth limit on the traversal and instead send more data than required, but it is unlikely to be sufficient. We expect that to make the selection more efficient, at a minimum, we would have to use a different domain decomposition method. Adopting the orthogonal bisection method [59], instead of the Peano-Hilbert curve, would allow us to significantly speed up the selection of particles that are within the search radius of the domain boundaries. This would result in a considerable speed up for the multi-GPU implementation of the code as the CPU will be less constrained. In this work we focus on single GPU performance and correctness and therefore leave this multi-GPU improvement for a future version.

## 4 Results

To validate the code and show the conservation properties we use a number of well-known SPH tests. We compare our results with the analytic solutions as well as with the results of PHANTOM [17]. For all tests we use the PHANTOM setup programs to generate the initial conditions.

Because BONSAI-SPH only support a global, constant, artificial viscosity we changed the default settings of PHANTOM to match this. For some of the tests this lead to quantitatively slightly different results when compared to those published in [17]. Unless indicated otherwise we use the $M_4$ cubic spline kernel, the Courant time-step parameter $C_{\text{cour}} = 0.3$, an adiabatic index of $\gamma = \frac{5}{3}$, and the viscosity switches $\alpha^{\text{AV}} = 1$, $\beta^{\text{AV}} = 2$ and $\alpha_u = 1$. Following the standard SPH validation tests we further show the scaling performance and energy conservation properties of BONSAI-SPH.

We used SPLASH [60] for plotting and extracting the exact solutions where applicable.

The computing system we used is an IBM S822LC (Minsky) system. This machine has two Power8 CPUs and 4 NVIDIA Tesla P100 GPUs. The Power8 CPUs have 8 cores, and each core can handle 8 threads. This gives a total of 128 threads that can be concurrently active. The operating system used is Red Hat 7.3, combined with CUDA 9.1.

### 4.1 Sod Shock Tube

Our first test is the standard Sod shock tube test [61]. Here we configure two different fluid states (left and right) with an initial discontinuity between the two states at $x = 0$. The left state ($x \leq 0$) has $[\rho, P] = [1, 1]$ with $256 \times 24 \times 24$ particles, while for the right state ($x > 0$)

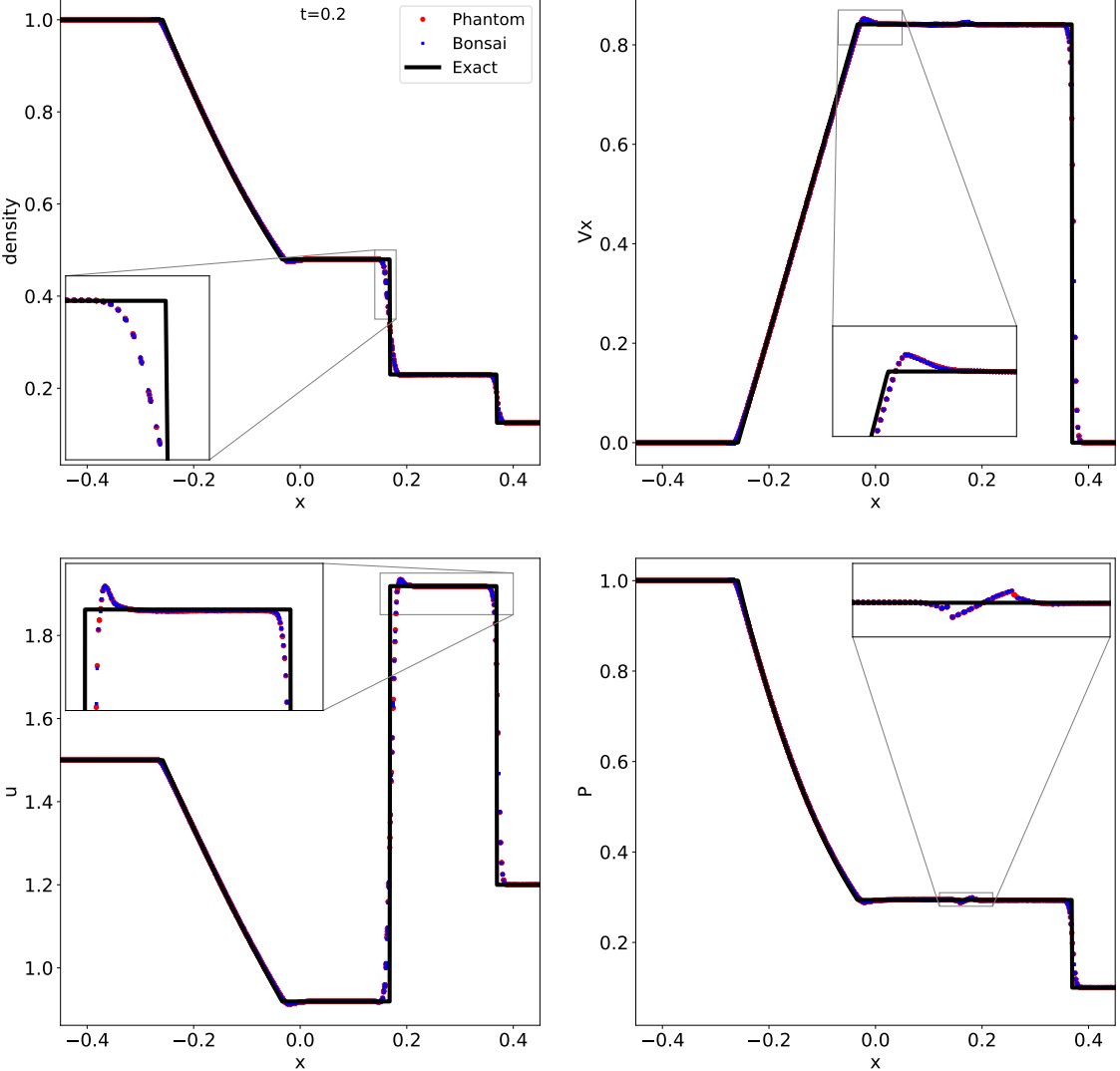

Figure 2: Sod shock test. Plotted are BONSAI-SPH (blue dots), PHANTOM (red dots), and the exact solution (black). From the top left to bottom right the panels show the density, velocity in the $x$-direction, energy and pressure plotted against the $x$-position. The results are shown for $t = 0.2$.

we have $[\rho, P] = [0.125, 0.1]$ with $128 \times 12 \times 12$ particles. For this test the $M_6$ quintic spline kernel is used (rather than the standard $M_4$ setting) and periodic boundaries for the $y$ and $z$ axes. Details on how the 3D initial conditions are generated can be found in [17].

In Fig. 2 we present the results at $t = 0.1$ for BONSAI-SPH, PHANTOM and the exact solution. The BONSAI-SPH and PHANTOM results are qualitatively indistinguishable.

## 4.2 Blast wave

As a second test we perform the blast wave test [53], which is more sensitive to implementation details as the differences between the left and right states is much larger. Here $[\rho, P] = [1, 1000]$ for $x \leq 0$ with $400 \times 12 \times 12$ particles and $[\rho, P] = [1.0, 0.1]$ with $400 \times 12 \times 2$ particles for $x > 0$. For this simulation we set $\gamma = \frac{7}{5}$, while the same viscosity and kernel settings as with the sod shock tube test are used. The results are presented in Fig. 3, where the exact solution is indicated with the solid line and the BONSAI-SPH and PHANTOM results are

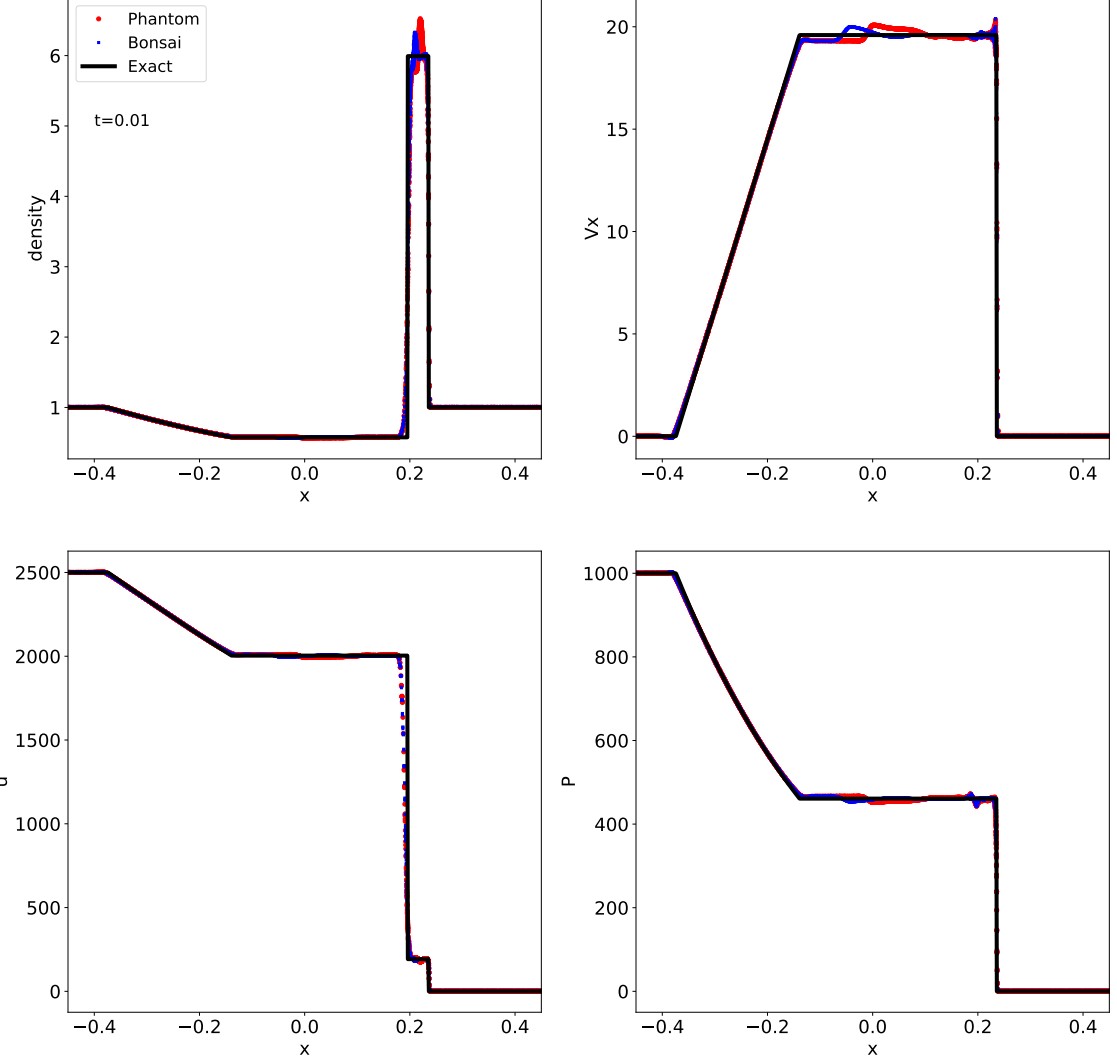

Figure 3: Blast wave test. Plotted are BONSAI-SPH (blue dots), PHANTOM (red dots), and the exact solution (black). From the top left to bottom right the panels show the density, velocity in the $x$-direction, energy and pressure plotted against the $x$-position. The results are shown for $t = 0.01$.

presented with the symbols. As with Fig. 2 the results between the exact solution and those of the simulations are comparable with the exception of small quantitative differences in the density and velocity profiles along the contact discontinuity at $x = 0.21$. There appears to be a minor phase difference between PHANTOM and BONSAI-SPH, but both codes show similar behaviour and the same error range. The phase difference is caused by the time-stepping method, but when we adopt a smaller value for $C_{cour}$ the phase difference is reduced.

## 4.3 Sedov blast wave

The Sedov-Taylor blast-wave test [62] can be compared to an analytic solution. This test follows the propagation of a blast wave in a spherical medium, and is often used to estimate the effect of supernovae explosions. We configure a uniform 3D box in which we place, at the center, a sphere composed of $100^3$ particles. The center particles are given a high initial energy which causes the explosive blast once the simulation is started. The results are presented in Fig. 4 and show BONSAI-SPH, PHANTOM and the analytic solution. Both the simulation codes

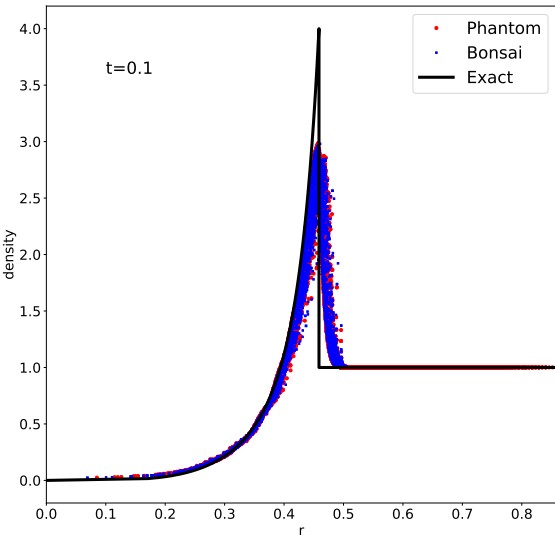

Figure 4: Sedov-Taylor Blast wave. Plotted are BONSAI-SPH (blue dots), PHANTOM (red dots), and the exact solution (solid line). The results are shown for $t = 0.1$.

fail to resolve the peak density that is predicted from the analytic solution, but other than that the results are consistent with the prediction.

## 4.4 Kelvin-Helmholtz instability

The Kelvin-Helmholtz (KH) instability test demonstrates the mixing behavior of two fluids with different densities at the moment the instability sets in [63, 64]. Much has been written about this test in particular with respect to the differences between SPH and grid codes. Traditionally SPH codes were unable to properly resolve this instability, but the addition of artificial viscosity and conductivity helped to resolve this [21]. Furthermore, the initial conditions have to be generated properly for a fair comparison between the various methods to simulate fluid dynamics [65]. In this work we use the method described in [65] which is implemented in the initial conditions generator of [17]. The KH test is in two dimensions but given that the code operates using three-dimensional coordinates it is executed as a flat bar. The box coordinates are between 0 and 1 in the $x$ and $y$ direction. Details on generating the initial conditions can be found in ( [17], section 5.1.4). The adiabatic index of the simulated fluid is $\gamma = \frac{5}{3}$, we use the $M_4$ cubic spline kernel and the default settings for the viscosity switches.

Since there is no analytic solution, the only way to validate our results (apart from energy conservation tests) is to compare it with previous implementations. We therefore ran the same initial conditions with PHANTOM, where we configured the free parameters to match those of BONSAI-SPH. For all the figures in this section we show the cross section of the particle density at $z = 0$.

In Fig. 5 we present a similar figure as presented in [17, 65] which shows the development of the instabilities as computed using BONSAI-SPH for 5 different resolutions ($n_x$=64, 128, 256, 512 and 1024) between $t = 0.5$ and $t = 2$. In between the BONSAI-SPH results we show the $n_x$=256 result as computed using PHANTOM. The results are qualitatively indistinguishable and both codes show the same behavior until the end of the simulation at $t = 10$ (not shown here).

At higher resolutions we see some noise appearing in the cross sections and the highest density contrasts become somewhat fuzzy. We therefore repeat the simulations using the

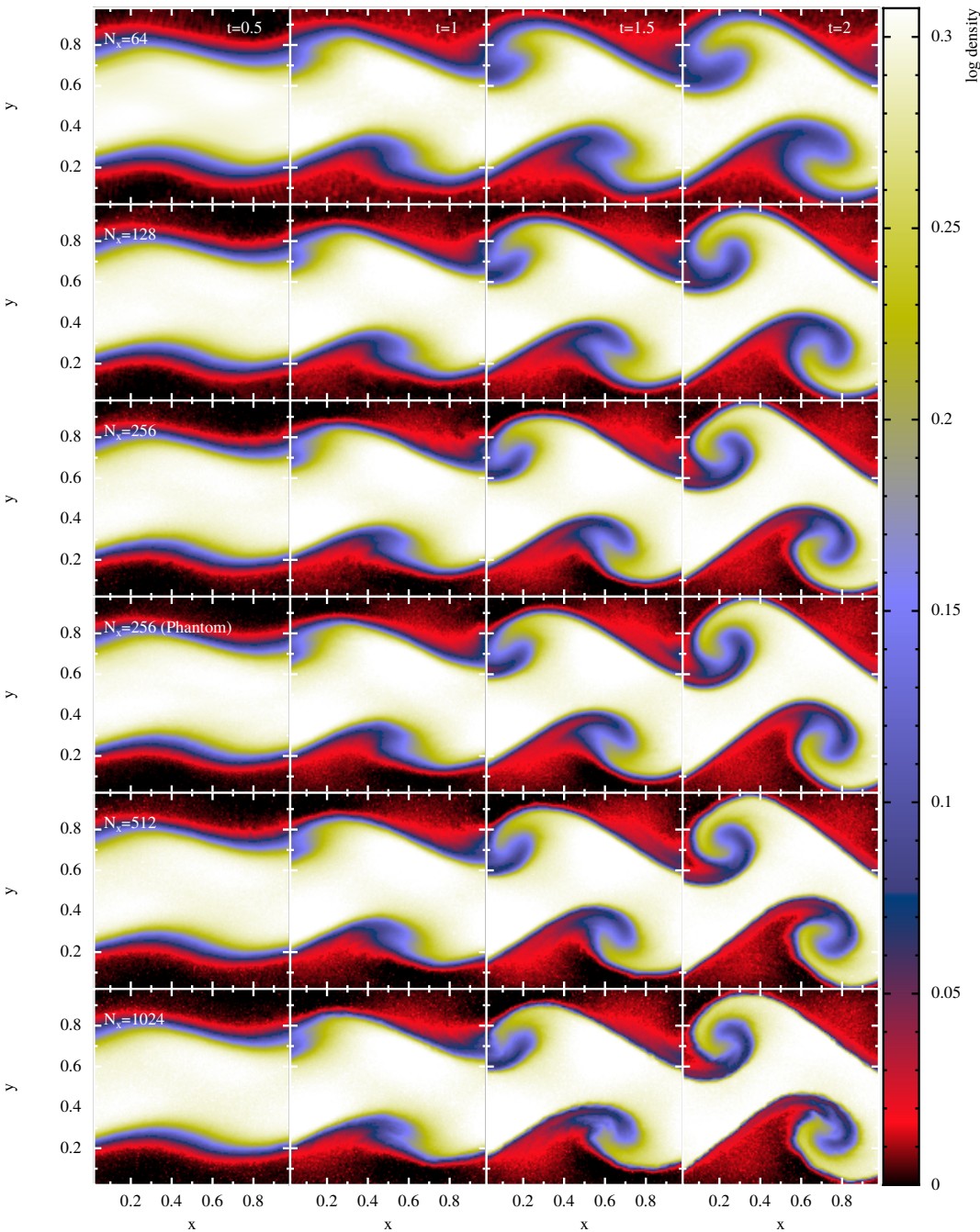

Figure 5: Kelvin-Helmholtz instability test. Presented are 5 different resolutions ($N_x$=64, 128, 256, 512 and 1024) where each column presents a different time-step ($t = 0.5$, 1.0, 1.5 and 2.0). The first 3 and final 2 rows are results from BONSAI-SPH the fourth row is generated using PHANTOM. The visualizations are the cross sections at $z = 0$.

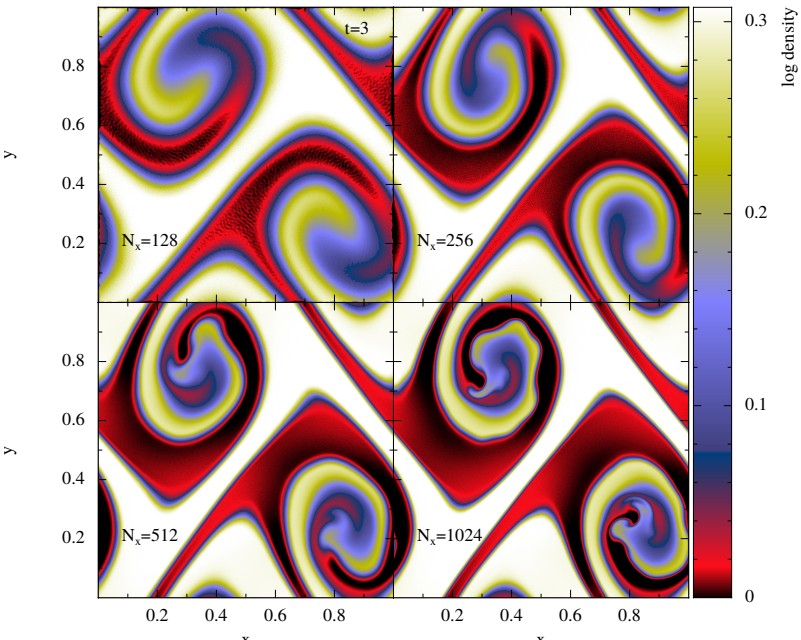

Figure 6: Kelvin-Helmholtz instability test. Presented are 4 different resolutions ($N_x$=128, 256, 512 and 1024) where each resolution occupies one of the panels. Results are generated using BONSAI-SPH with the $M_6$ quintic kernel. Note this is at $t = 3$ while Fig. 5 is at $t = 2$.

higher order $M_6$ quintic kernel. The higher order kernel should give smoother results because more neighbours are involved. This time we used 4 different resolutions ($n_x$=128, 256, 512 and 1024) and compared the $t = 3$ snapshot. The results are presented in Fig. 6, in a manner similar to the results from ENZO presented in [65][Fig. 8]. Both our figures use the same colour scale.

The results are much smoother than the results from Fig. 5 for $n_x$=512 and 1024. This demonstrates that BONSAI-SPH behaves as expected and is able to resolve the tiny features required to generate mixing.

## 4.5 Energy conservation

In the previous sections we used the analytic solution to validate the code, alternatively it is also possible to keep track of the energy conservation to verify that the code behaves correctly.

We selected three simulations from the previous sections and extracted the energy conservation over the course of the simulation for both PHANTOM and BONSAI-SPH. We selected the *Sod Shock Tube*, the *Sedov blast Wave*, and the *Kelvin-Helmholtz instability* run with $N_x$=256 using the $M_6$ kernel.

The comparison is presented in Fig. 7, and computed using,

$$dE = (E_0 - E_t)/E_0 \, . \tag{16}$$

In the left panel the results of the *Sod Shock Tube* are presented and both BONSAI-SPH and PHANTOM show similar behaviour where both codes give an energy error on the order of $10^{-6}$.

The middle panel shows the *Sedov blast Wave* test. Using the default time-step parameter we found that the energy error of BONSAI-SPH in the first few steps behaves quite erratically, we therefore ran two more tests where we decreased the $C_{\mathrm{cour}}$ value (Eq. 8). This stabilized the results, and brought it more in line with the results of PHANTOM which, in contrast to

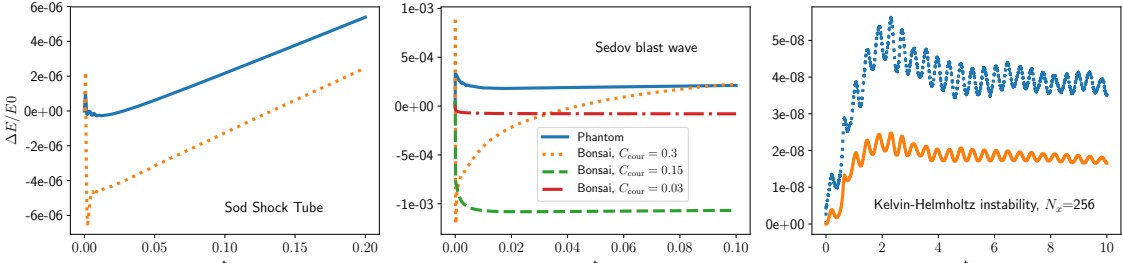

Figure 7: Energy error of BONSAI-SPH vs PHANTOM for the *Sod Shock Tube* (left panel), *Sedov blast Wave* (middle panel) and *Kelvin-Helmholtz instability* (right panel). In all panels the $x$-axis indicates the time since the start of the simulation and the $y$-axis the energy error via Eq. 16. For all panels, the solid line shows the results as obtained with PHANTOM and the dotted line the results of BONSAI-SPH using the default time-step. In addition, the middle panel shows the results of BONSAI-SPH using $C_{cour}$=0.15 (dashed-line) and $C_{cour}$=0.03 (dash-dotted line).

BONSAI-SPH uses a combination of 6 different time-step criteria to determine the step being used.

The right panel shows the *Kelvin-Helmholtz instability* data. The result of BONSAI-SPH is slightly worse than that of PHANTOM, but is within the same order of magnitude and shows similar behaviour.

The main reason, apart from the time-step method mentioned above, for the difference between the two codes is the used numerical precision. BONSAI-SPH uses `float32` whereas PHANTOM uses the `float64` data-type. This higher precision improves the accuracy and reduces the noise in the computations.

## 4.6 Performance

One of our goals of developing BONSAI-SPH was to get access to a faster SPH code by taking advantage of the GPU's computational resources. In this subsection we therefore compare the performance of BONSAI-SPH with that of PHANTOM. For BONSAI-SPH we used a single P100 GPU and two CPU threads. We use one thread for controlling the GPU and the other thread for writing data, no further threads are required as we do not use multiple GPUs, nor do we do any post-processing. We used the following properties for the tree-structure, $N_{leaf} = 16$, $N_{crit} = 8$, no further tuning is required to run BONSAI-SPH[3].

As mentioned at the beginning of this section the used `Power8` CPU is capable of running 64 threads per CPU. These threads, however, do share some of the hardware resources and therefore the ideal number of threads differs per application. Given that PHANTOM is capable of making optimal use of multiple threads [17] we had to find the most optimal number of threads to make a fair comparison between both codes. For this we ran a set of Kelvin Helmholtz test calculations to determine the optimum number of CPU threads. The results of this experiment is presented in Fig. 8. The speed-up indicates that it is beneficial to add additional CPU threads, until the peak is reached for $N_{thread} = 64$. Using more than 64 threads does not lead to better performance because the threads are competing for the same resources. We therefore set the number of CPU threads used by PHANTOM to 64.

Next, we compare the performance of BONSAI-SPH with respect to PHANTOM. For this we use the KH simulations. We chose this test because, of all our models these simulations contained the most particles and has a wide range of density contrasts. For the reasons described

---

[3]for details see [38]

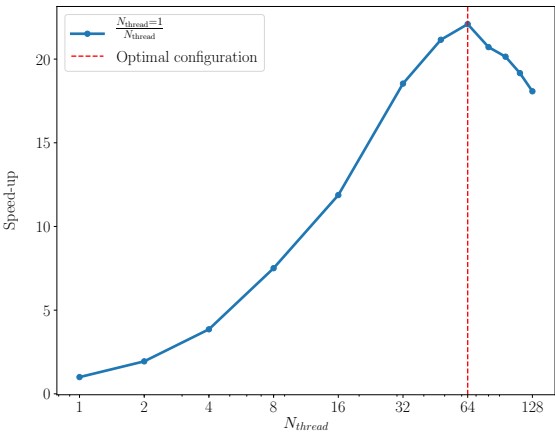

Figure 8: Effect of increasing the number of compute threads used by PHANTOM on the execution speed. The $x$-axis shows the number of `OpenMP` threads and the $y$-axis the speed-up compared to single thread execution. The relative speed is indicated by the solid blue line, the best performance is reached when $N_{thread}$=64, indicated by the vertical red dashed line. Data generated using the $N_x$=128 Kelvin-Helmholtz dataset.

above we use $N_{thread}$=64 for PHANTOM and compare that to the single GPU timing results of BONSAI-SPH.

Because BONSAI-SPH performs fewer time-steps per simulation than PHANTOM we base our performance comparison on the average wall-clock time *per simulation step* instead of the total wall-clock time. To ensure that reported numbers are stable, and based on enough data points, we execute the KH simulation for 10 time-units (2 units for $N_{512}$ and $N_{1024}$). This results in thousands of time-steps per simulation. We verified that the time-per-step is roughly constant over the course of the simulation to ensure that the reported comparison is valid for the whole simulation. We furthermore performed multiple independent simulations for the $N_{64}$ configuration to verify that the timing data between runs is consistent. In all cases we found that there is little to no variation between runs and over the course of the run so we only present the average data of a single run. This also allowed us to simulate a shorter time-frame for the large $N$ models in order to get the results within a day instead of a month.

In order to make a proper performance comparison between BONSAI-SPH and PHANTOM we have to take into account the numerical precision difference we mentioned earlier. Therefore we did the performance evaluation using two different versions of PHANTOM. The first version uses the default compiler settings which results in double precision (64bit) floating point operations. For the second version we modified the compiler flags[4] to build a version that only uses 32bit floating point operations, e.g. the same accuracy as BONSAI-SPH.

To compute the speed-up we divide the averaged time per simulation step of PHANTOM with that of BONSAI-SPH the results are presented in Fig. 9. We see that BONSAI-SPH is a factor 4 to 10 times faster than PHANTOM. Where the speed-up is smaller for smaller datasets, which can easily be explained by the fact that for smaller dataset sizes the GPU is underutilized. For our largest dataset size (14M particles) BONSAI-SPH is almost a factor 10 faster than PHANTOM when using 64bit computations and a factor 8.6 faster when PHANTOM is using 32bit computations. The minor difference between the 64bit and 32bit versions suggest that the performance difference between those two compute modes on the Power8 architecture is minor, especially when there is no explicit usage of the vector assembly instructions in

---

[4]specifically the `"DOUBLEPRECISION"` flag, used to disable 64bit computations.

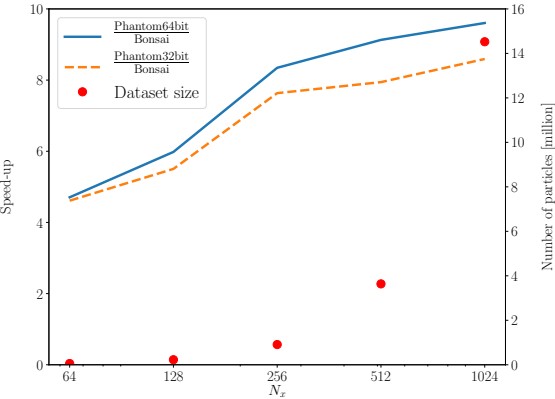

Figure 9: Speed-up when using BONSAI-SPH vs PHANTOM. The $x$-axis indicates the Kelvin-Helmholtz dataset size, the left $y$-axis indicates the speed-up. The red circles (right $y$-axis) indicate the number of particles in the model. The solid blue (dashed orange) line indicates the difference between BONSAI-SPH and the 64bit (32bit) version of PHANTOM. The scaling data is obtained by running the same model using both simulation codes for the same amount of simulation time and then comparing the average time per iteration step.

the source code.

## 5   Conclusions

In this work we introduced a new GPU accelerated SPH code called BONSAI-SPH. Our objective was to develop a GPU optimized solver for fluid dynamics. We demonstrate that BONSAI-SPH can compete in terms of precision and accuracy with state-of-the-art codes when simulating fluids using the modern SPH equations. Not only do the results match, BONSAI-SPH also executes them up to a factor 10 faster. This enables researchers to do more or larger simulations in the same wall-clock time-frame. In the same way as we developed the BONSAI pure gravitational code, we hope that this allows researchers to perform simulations using resolutions that where hitherto beyond reach of modern computers, such as the ones presented in [40].

However, we did not develop BONSAI-SPH as a replacement for stand alone SPH codes. Not all the features, such as sub-grid physics, that codes such as Gadget [19] and PHANTOM offer are implemented. We specifically focused on implementing the fundamental features that allow faster exploration of parameter space. Once a final configuration has been found users can opt to run that setting with a slower, but more versatile code. Alternatively the user can combine BONSAI-SPH with other features via the AMUSE framework [43, 66–68]. This allows fast prototyping while still benefiting from the fast execution of the density and force computations.

Future work that we plan ourselves are the optimization of the multi-GPU code path. Currently the code is able to take advantage of multiple-GPUs, but there is barely and performance improvement because the CPU part of the code slows the calculation down. However, if one wants to run models that do not fit in the memory of a single GPU then this can be done already with the currently published version. Other relatively easy features that could be added are additional time-step switches, stopping conditions, or a more optimal, per particle group, determination of the number of density iterations required.

## Acknowledgments

We thank Terrence Tricco for discussions and Daniel Price and collaborators for making PHAN-TOM open source which helped us during and after the development of BONSAI-SPH to verify the correctness of our implementation. We further like to thank Evghenii Gaburov, Inti Pelupessy and Natsuki Hosono for discussions and advice. This work was supported by the Netherlands Research School for Astronomy (NOVA), NWO (grant # 621.016.701 [LGM-II]) and by the European Union's Horizon 2020 research and innovation program under grant agreement No 671564 (COMPAT project).

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
