# Peer review of "Bonsai-SPH: A GPU accelerated astrophysical Smoothed Particle Hydrodynamics code"

_SciPost Physics, doi:SciPost Astro. 1, 001 (2020)_

## Round 1 · Referee Report · Mark Vogelsberger (Referee 1) · 2020-1-1

Strengths

1 - The paper presents a very clear description of the SPH implementation in Bonsai-SPH.

2 - The methods and test problems are clearly laid out.

3 - The figures are very polished.

Weaknesses

1 - It would be nice to see some small scientific application problems in the paper beyond the test problems. Maybe the authors can add something along these lines?

2 - The paper does not detail how the code performs on parallel problems. Can the authors add a figure demonstrating well parallel processing works on multiple GPUs?

3 - The last years have seen various novel SPH implementations. It would be good if the introduction could discuss some of those. The authors should also discuss a bit more in detail the advantages and disadvantages of SPH compared to other techniques.

Report

The paper "Bonsai-SPH: A GPU accelerated astrophysical Smoothed
Particle Hydrodynamics code" by Bedorf and Zwart presents the smoothed-particle hydrodynamics simulation code, Bonsai-SPH,
which is a continuation of the gravity-only hierarchical N-body code Bonsai. The code is optimized for GPU accelerators. The authors demonstrate that Bonsai-SPH produces simulation results comparable
with state-of-the-art, CPU based, codes, but using an order of magnitude less
computation time.

Overall this paper presents a concise and clear description of the methods and test problems. There are a few points where the authors could expand the paper slightly. I discuss those under 'weaknesses'. It would be nice to see some of these suggestions being implemented by the authors. However, this is not a requirement since the paper is in its current form already publishable.

Requested changes

1 - Add some small scientific application at the end that goes beyond the pure test problems.

2 - Discuss in more depth the parallel aspects of the code.

3 - Extend the intro by a more detailed review of recent efforts in SPH implementations and also discuss in more detail the advantages and disadvantages of SPH.

---

## Round 2 · Author Response

We have updated the manuscript to take into the comments from the referee, for changes see the 'List of changes' section. The application example we will leave for a follow up paper.

---

## Round 2 · List of Changes

Changes made:

Introduction
* Expanded upon differences/advantages/disadvantages between SPH and Grid codes
* Added a paragraph about moving grid codes that combine advantages of SPH and grid codes into a single method.

Multi-GPU section
* Expanded upon the problems faced
* Explain in further detail the performance difference between the multi-GPU gravity and multi-GPU SPH methods and which steps have to be taken to optimize the SPH method.

You are currently on this page

Resubmission 1909.07439v2 on 28 February 2020

---

## Editorial Decision

published